# Evident: a Development Methodology and a Knowledge Base Topology for Data Mining, Machine Learning and General Knowledge Management

## Abstract

Software has been developed for knowledge discovery, prediction and management for over 30 years. However, there are still unresolved pain points when using existing project development and artifact management methodologies. Historically, there has been a lack of applicable methodologies. Further, methodologies that have been applied, such as Agile, have several limitations including scientific unfalsifiability that reduce their applicability. *Evident*, a development methodology rooted in the philosophy of logical reasoning and *EKB*, a knowledge base topology, are proposed. Many pain points in data mining, machine learning and general knowledge management are alleviated conceptually. *Evident* can be extended potentially to accelerate philosophical exploration, science discovery, education as well as knowledge sharing & retention across the globe. *EKB* offers one solution of storing information as knowledge, a granular level above data. Related topics in computer history, software engineering, database, sensing hardware, philosophy, and project & organization & military managements are also discussed.

## 1 Introduction

Necessity is the mother of invention, claimed Plato [1]. Deficient in rigorous scientific scrutinization as the statement is, major methodology evolutions in software development did not emerge until the emergence of major computer innovations and thereafter elevated effort orchestration needs.

In the 1940s, digital programmable electronic computers revolutionized scientific calculation done previously with mechanical and analog computing machines [2]. Assembly (1947) [3, 4] and high level (1953) [5, 6] programming languages rose to harness the unprecedented and ever-increasing computing power, Eventually the term software was coined (1953) [7]. Two Software Development Methodologies (SDMs) were proposed: 1. a project breakdown of sequential phases, the essence of Waterfall (see Fig 1a), first presented no later than 1956 [8] and 2. iterative and incremental SDM, the essence of Agile (see Fib 1b), first executed no later than 1957 [9].

In the 1960s, operating systems (1962) emerged to orchestrate multiple computation tasks[10]. This signified the shift of computer development from single-task specialized machines for military and academia to machines accessible to the general public. The shift was exemplified by The Mother of All Demos (1968) which demonstrated many fundamental elements of personal computing for the first time [11] and showed how software had evolved in both diversity and complexity. Meanwhile, the first formal detailed diagram of the Waterfall methodology appeared in literature (See Fig 1a) (1970) [12] and the name of Waterfall was ultimately coined (1976) [13]. Agile variants such as evolutionary project management [14] and adaptive SDM [15] appeared in the early 1970s, although no clear preference between Waterfall and Agile variants was found in the literature.

Submitted to 36th Conference on Neural Information Processing Systems (NeurIPS 2022). Do not distribute.

Table 1: Major Software Usage Evolutions and Methodology Developments

| Period | Technology | Software Usage | Major Methodology Development |
|---|---|---|---|
| 1940s | Programing Language | Scientific Calculation | First Waterfall variant presentation (1956) [8]; first Agile variant execution (1957)[9]. |
| 1960s | Operating System | Shifting to Applications | First detailed diagram of Waterfall idea(1970)[12], Waterfall name (1976)[13]; Agile variants: evolutionary project management [14] & adaptive SDM [15](early 1970s). |
| 1980s | GUI & Internet | PC & Internet Applications | Waterfall standardized in military (1985)[20]; The Manifesto signed (2001)[24]. Agile significantly more popular than Waterfall. |
| Around 1990 | Data Storage | Knowledge Discovery, Prediction and Management | NA |

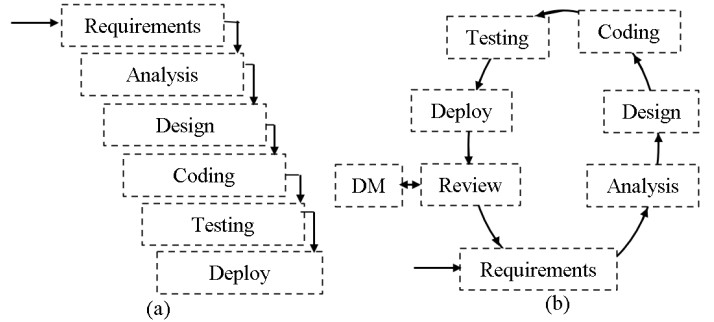

Fig 1 a) A typical Waterfall design diagram [12] b) An iterative, evolutionary and incremental design cycle commonly viewed as Agile[9] and how DM can improve each design cycle by discovering Knowledge about user needs during Review.

Fig 2. Relations among DM, ML and KM for their pain points

In the 1980s, personal computers entered households[16] followed by graphic user interface (GUI) (1983) [16]. The Internet Protocol Suite (TCP/IP) was standardized (1982)[17] and commercial Internet service providers emerged (1989) [18, 19] Unprecedented user-computer interactions and user-user communications created tremendous software needs, while Waterfall was still widely deployed in software development. United States Department of Defense issued a military standard describing Waterfall as the required military software development process (1985) [20]. However, software user needs grew so fast that, the heavy Waterfall SDM failed to deliver in pace. Consequently, a number of light weight SDMs were proposed and practiced (1990s) [4, 21, 22, 23]. Eventually The Manifesto for Agile Software Development (The Manifesto) [24] was signed by 17 practitioners of light-weight SDM (2001) and became the de facto SDM.

Around 1990, data storage capacities grew significantly and software usages in Data Mining (DM, defined as knowledge discovery from data) reached the tipping point. While data and software's storage manners differ between Von Neumann and Harvard architectures, data storage capacity growth empowered software to discover knowledge supported by scientific evidence (defined as Knowledge) that people never had access to. Corporations started to analyze customers' behavior and make business decisions based on Knowledge (1990s) [25]. The first DM methodology, Cross-Industry Standard Process for Data Mining (CRISP-DM) was conceived (1996) [26, 27]. However, CRISP-DM and its variants appear more of a theoretical framework, offer little meaningful or actionable guidance, and therefore have not gotten much traction. In addition, CRISP-DM is concerned only with DM, not Machine Learning (ML, defined as to deliver an algorithm (Algo) for Knowledge prediction) or Knowledge Management (KM) in general for science, medicine, military and so on. Due to the absence of alternative methodologies(see Table 1), Agile is still being offered up for DM, ML, and KM [28, 29] with questions being asked about its appropriateness [30, 31].

This paper discusses limitations in Agile as a scientific claim and why it may not address the current pain points of DM, ML and KM, which are later summarized. *Evident* along with *Evident Knowledge Base* (*EKB*) is proposed as a project development and artifact management methodology. *Evident*'s potential in alleviating many current pain points is demonstrated conceptually. Unalleviated pain

points and future work to fulfill the potential are also discussed. Beyond software development, *Evident* is illustrated to be applicable in many aspects of society. *EKB* is demonstrated as one potential infrastructure to store information as Knowledge, a granular level above data.

# 2 Agile ambiguity and unfasifiability

Most people regard Agile as iterative, evolutionary and incremental software development [9] (see Fig 1b) and many claim to be Agile practitioners. However, Agile empirical evidence is mixed and hard to find [32, 30] while no measurable scientific evidence has been found at all. Although control experiment challenges or absence of quantitative project Agility measurements may explain no measurable scientific evidence, concerns remain with the ambiguity with which Agile's approaches and scope are defined in The Manifesto.

## 2.1 Agile approaches are vaguely defined in The Manifesto

The Manifesto includes 4 values and 12 principles [24]. The goal is crystal clear: to rapidly deliver quality software that meets user needs, but not so much can be found for how to get there. Most of the Values and Principles appear to be goals but not approaches (see Appendix); some are concerned with approaches but vaguely defined; only four principles are actionable, which turn out to have no relevance in how to implement iterative, evolutionary or incremental development. Agile Alliance, co-founded by some original signers of The Manifesto, defines Agile as "an umbrella term for a set of frameworks and practices" from which Agile practitioners "figure out the right things to do given your particular context." [33] Unfortunately, no actionable approaches are defined.

Therefore although there are numerous frameworks under the Agile umbrella [34, 35], it's impossible to determine if a development practice is Agile and the claim of Agile practice becomes unfalsifiable. Because falsifiability is the standard evaluating scientific against non-scientific claims introduced by Karl Popper [36], Agile is not a scientific claim. Consequently, no observable scientific evidence can prove or disprove Agile, because technically no one can determine if a project is Agile or not in the first place. If an Agile rollout "fails", Agile proponents can always argue that the Agile rollout was not implemented correctly.

Meanwhile Agile practitioners cannot determine if they practice Agile correctly either. Projects employing Agile frameworks such as Test Driven Development or Feature Driven Development may not even realize that the projects may not be adaptive to new user needs. People who are essentially practicing Waterfall may believe they are practicing Agile only because they implement each sequential Waterfall phase incrementally or simply use Scrum or Kanban.

In defense, some proponents claim Agile as a philosophy [37, 38]. Granted Agile's goal may fit into Axiology, one of Philosophy's four domains (the rest as Metaphysics, Epistemology and Logic) [39], concerned with what is good, it appears to be a common understanding and offers little value when Agile's approaches are vaguely defined.

## 2.2 No scopes are defined in The Manifesto

"The right things to do given your particular context" by Agile Alliance [33] are expected to be found within Agile's frameworks, otherwise Agile is not practiced right. With no scope defined, Agile seems to cover the scope of all softwares. However, some softwares have non-incremental needs or simply only one need, e.g. to solve one specific partial differential equation numerically. Their needs are either met or not at all. No iteration or evolutionary Agile design cycles exist.

Agile is also not applicable for Knowledge discovery tasks such as DM. In a typical Agile development cycle (Fig 1b), Review phase is to discover Knowledge about user needs, which can be done through DM. Therefore Agile should not be applicable to DM, one phase of its own design cycle.

Table 2: Pain Points of DM, ML and KM.

| Pain Points for DM | | |
|---|---|---|
| | Struggles to deliver fast with technical debt | abc |
| | Project Progress not easily measurable | b |
| | Uncertainty in project timeline | b |
| Overall | Activities not easily trackable or reproducible | bc |

| | | |
|---|---|---|
| | Not scalable in terms of both collaborator number and project maintenance | ab |
| | Few general project design patterns | abc |
| | Anti-patterns not uncommon | abc |
| Collabor-ation | No methodologies to orchestrate team of size commonly seen in software development | b |
| | Few intuitive manners to divide task among team members | ab |
| | Deficient common awareness in needs for process improvement | a |
| | Tasks completed or ideas explored by team members cannot be easily found and reproduced causing duplicated work. | bc |
| Data | Data compromised in availability, accuracy and consistency during acquisition | x |
| | Data preprocessing process not standardized such as data labeling, object detection (e.g. identify object pixels in images) causing unstable data dependency, cascade correction and uncertainty in project progress | b |
| | Underutilized data may take unnecessary resource | b |
| | Data may be presented in different data type such as integer, float or string, causing unnecessary data dependency for Algo and experiments | b |
| Knowledge Discovery / ML Algo Research | Off-the-shelf models are available for DM automation. However, DM automation has not become a common practice. | abc |
| | Inefficient in-house model code implementation not uncommon | b |
| | No appropriate version control tools. Current version control tools such as git are designed to only keep the best version Algo/experiment available, while DM and ML need multiple versions available concurrently for reference | abc |
| | No easy solution to request flexible data storage, memory and computation capacity as needed. Hard drive, RAM, CPU and GPU are difficult to allocate even on the cloud. | x |
| | The use of other Algos' output as input results in correction cascades | b |
| | Algo is a sequential computation process different from typical software applications with a number of independent features. Difficult to assign one Algo development into multiple team members | x |
| | Algo user may have no clear understanding about the Algo and deploys it outside its scope. | b |
| | Multiple programing language smell | x |
| Pain Points for ML | | |
| Algo Production | Significant efforts of research Algo migration into production | bc |
| | Even more significant efforts if production Algo is written in a different language than the language used in research, e.g. in embedded system | x |
| | For Algo analyzing sensor data such as cameras or bio-sensors, product grade data won't be available for Algo research until sensor hardware designs are complete. Algo becomes the product release bottleneck, resulting in either sub-optimal production Algo or delayed product release. | x |
| | Production data source is inconsistent with research data source | x |
| | Algo production is often done by team members, most likely software engineers, who did not produce the Algo, causing misuse | abc |
| | Algo needs to load data in real time during production but most often not in real time during research, causing unnecessary Algo code re-factoring | b |
| | Prototype Algo may be accidentally run in production causing damages | x |
| | No straightforward way to organize codes repository for research and production team members work in the same repository | abc |
| | Dead code path | b |
| Feedback Loop | Algo update workflow not straight forward | b |
| | Few clear pattern designs for monitoring Algo performance in production | b |
| | Actions based on unseen data predicted by Algo may alter observed data | x |
| Pain Points for KM | | |
| Management Tool | Few tools or resource help people check if Knowledge formed is well supported by evidence, especially when evidence appears long after presumed Knowledge has been formed. | abc |
| | Knowledge dissemination among community has always been a challenge. | abc |
| | Knowledges formed by different organizations are not easy to combine | abc |

| | Knowledge formed within organizations is not easy to share and retain. | abc |
| Standardization | Knowledge has been recorded in sentences or articles. Few standardized ways to represent general Knowledge. | abc |

a/b/c: Pain points that can be alleviated by *Evident*'s character a, b or c. x: Pain points that cannot be alleviated by *Evident*.

# 3 Pain points for DM, ML and KM

Owing to the absence of applicable methodologies, pain points have been continuously reported for DM, ML and KM [40, 41, 42, 43] (see Table 2) in the current big data era with explosive growth in data volume, variety and velocity.

DM & ML's Algo typically comprises a data computation flow (defined as a Model, supervised or unsupervised), such as logistic regression, and its configuration, such as logistic regression coefficients. DM employs off-the-shelf or in-house Models to discover Knowledge from data. ML compares Knowledges discovered by DM by candidate models and deploys the one that performs best with its configuration, as the production Algo, for Knowledge prediction in production. Therefore, DM's pain points still apply to ML. Meanwhile because DM and ML are special forms of KM, their pain points are also applicable for KM (see Fig 2).

Generally speaking, DM has not been regarded highly collaborative and scalable activities to deliver high throughput Knowledge. It's rare to see hundreds of contributors in a DM project, unlike for example some complicated open source software projects that, deliver promptly, efficiently and continuously for years or even decades [44, 45]. It is also rare of mass Knowledge production in a organized and standardized manner with high production yield for a unit period, commonly seen in consumer products such as automobiles or toothpastes. DM often struggles to deliver Knowledge rapidly with technical debts in reproducibility, measurability, trackability. DM needs to not only handle artifacts of different modalities such as documents, codes and data, but also address computation and data storage resource requests potentially across multiple platforms. Raw or preprocessed data can be compromised in availability, accuracy and consistency. Data dependency and entangled models often cause cascaded correction and uncertainty in project planning. In addition, routine tasks such as quarterly or annual finance analysis mostly have not be automated. Tools and project management methodologies are highly in demand to fulfill DM's potential and deliver values.

In ML, the team members, usually software engineers or product managers, that deploy an Algo in production to predict future data may not have produced the Algo and may misuse it. The Algo codes are often refactored sometimes in different programming languages, operating systems or even in fixed point instead of float point. If the Algo is to be deployed on a data acquisition product such as cameras or bio-sensors, no product grade data are available for Algo research until the sensor hardware design is finalized to enable data collection. Algo research therefore becomes the bottleneck for product release. Frequently sub-optimal Algo is deployed to meet the deadline or projects become delayed. What's more, production data may come from different sources compared to research data potentially caused by, e.g. sensor upgrade or downgrade, resulting in the under performance of production Algo. After Algo deployment, no straightforward way exists to monitor Algo performance or thereafter update Algo. Future Knowledge predicated by the deployed Algo may encourage Algo users to alter decisions, which leads to the formation of future data with unanticipated hidden feedback loops. In addition, prototype Algo or dead codes may be accidentally run in production potentially causing catastrophic consequences.

Although scientific methods have guided people to discover Knowledge and improve practices such as evidence-based medicine [46] and experiment based military development [47], people still form Knowledge that lacks in supporting evidence [48]. One possible reason is the unavailability of tools or resources to check if the Knowledge formed is well supported by evidence, especially when there is a significant time gap between the Knowledge formed and the appearance of supporting evidence, e.g. for long-term investments or corporation strategies. Knowledge dissemination and retention are also huge challenges among the community and organization [49]. Furthermore, Knowledge is mostly recorded in the form of articles. However, because articles writing has not been and probably will never be standardized, Knowledge has not been able to be represented in a standardized manner for definition, reference and storage. The same Knowledge recorded in different sentences or even languages may be interpreted differently.

## 4 *Evident*: a project development and artifact management methodology

### 4.1 Definition and scope

*Evident* is a methodology of project, including but not limited to software, development and artifact management for DM, ML and KM, characterized by

   a. project development mimicking a continuous process of logical reasoning in philosophy;

   b. project activities or artifacts are broken into containers of Observations, Hypotheses and Tests (collectively defined as Containers);

   c. directional association constructions towards and only towards Test Containers to represent Knowledge.

Observation is a collection of facts. A Hypothesis is Knowledge to be formed out of Observation. A Test is a Hypothesis evaluation process using Observation to prove or disprove the Hypothesis with or without confidence levels. Containers indicate Observations, Hypotheses and Tests can only be added or removed as a block.

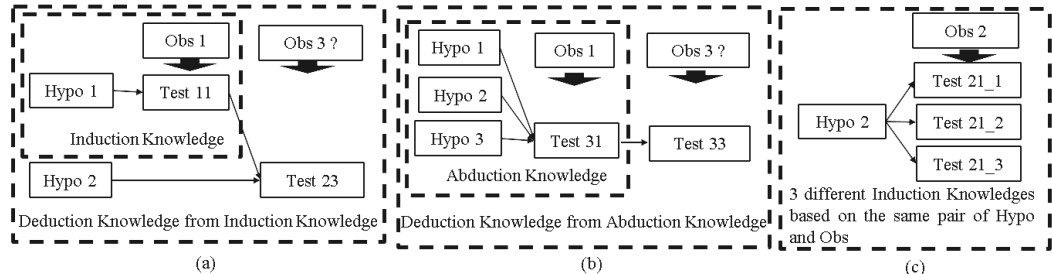

Fig 3 Induction, abduction and deduction Knowledge represented in *Evident* (Hypo: Hypothesis; Obs: Observation)

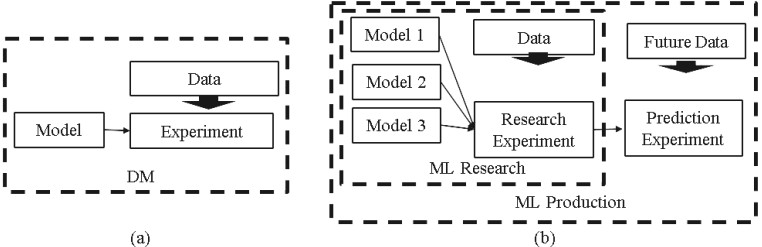

Fig 4 DM, ML processes represented in *Evident*. a) DM mimic Knowledge induction in Fig 3a; b) ML Research and Production mimic Knowledge abduction and Knowledge deduction in Fig 3b.

### 4.2 Knowledge representation

Loosely speaking, a Test associated with a Hypothesis and an Observation represents induction Knowledge (see Fig 3a); a Test associated with a Hypothesis set and an Observation represents abduction Knowledge (see Fig 3b), a Hypothesis associated Test that is also associated with an induction or abduction Knowledge Test represents deduction Knowledge or prediction (see Fig 3a&b). Deduction Knowledge becomes induction Knowledge once Observation proving or disproving deduction Knowledge is associated with Test, while stay deducted Knowledge if the associated Observation overlooks (fails to either prove or disprove), the deduction Knowledge. Multiple Tests associated with the same pair of Hypothesis(es) and Observation represent multiple different Knowledges based on different evaluation metrics (e.g. profit maximization or cost minimization) or Observation usage strategies (e.g. cross-validation grouping) (see Fig 3c).

DM may be regarded as Knowledge induction with data as Observation, model as Hypothesis to be evaluated and data analysis experiment as model test on data to form induction Knowledge with statistical confidence (see Fig 4a). Similarly, ML is Knowledge abduction. An example is a data analysis experiment that picks the best off-the-shelf or in-house model that best explains the data to form the Algo for production (see Fig 4b). Experiments employing different cost functions (e.g.

RMSE, AUC or correlation coefficients), statistical confidence levels or data allocation strategies for training and testing may result in different Knowledges or Algos.

## 4.3  Project development

*Evident* project developments are intuitively broken down into two granular levels: Knowledges and Containers. Mimicking logical reasoning in philosophy, each project period develops a batch of independent Knowledges or Containers assigned to teams of various sizes to maximize unit time throughput (see Fig 5). The next batches of Knowledge or Containers can be adaptively planned after period reviews or retrieved from backlogs. *Evident* is compatible with Kanban, Scrum or other development tools or frameworks for project planning and development of a single Knowledge or Container.Any tools or frameworks that do not compromise the project breakdown into Knowledge and Containers are applicable.

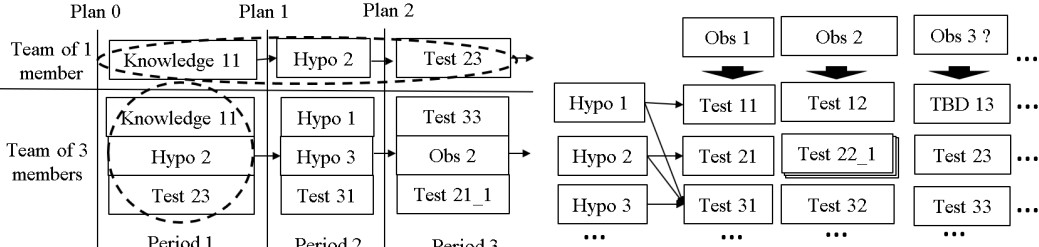

Fig. 5 One exemplary *Evident* project development process to produce Knowledges in Fig 3 a&b&c. Circled tasks may take 1 team member 3 periods, or take a team of 3 members 1 period. Knowledge 11 represents Obs 1 & Hypo 1 & Test 11 in Fig 3a.

Fig. 6 An exemplary EKB composed of Hypo, Obs and Test containers that can store induction, abduction and deduction Knowledges in Fig 3 a&b&c and project development artifacts in Fig 5 continuously.

## 4.4  Artifact management: *EKB*

*Evident* artifact management may build on a topology of a relational Knowledge base, named *EKB*, composed of *Evident* Containers and directional associations (see Fig 6). Knowledges can be reproduced by Containers stored in *EKB*. Oversimplified as a table, *EKB* columns represent Observations; rows represent Hypotheses; values represent Tests or Test to be done (TBD). A Test can only be associated with one Observation, even if the associated Observation overlooks the Hypothesis(es) associated with the Test. Any new Observation proving, disproving or overlooking the same Hypothesis(es) occupies a column in *EKB*.

### 4.4.1  *EKB* stores containers and Knowledges continuously

When new Hypotheses, Observations or deduction Knowledge Tests are developed, new rows or columns of TBDs are inserted. A Test associated with Hypothesis(es) and a Observation can be stored in the designated row and column to represent different Knowledges.

An induction Knowledge Test is placed in the row of its associated Hypothesis and the column of its associated Observation (see Fig 3a&c & Fig 6). An abduction Knowledge Test is placed in the row of the Hypothesis best explaining the Observation (see Fig 3b & Fig 6). A deduction Knowledge is placed in the row of its associated Hypothesis and the column of the pending Observation (see Fig 3a&b & Fig 6). Once an Observation becomes available proving or disproving the Hypothesis, a deduction Knowledge becomes an induction Knowledge. Multiple Tests representing different Knowledges can be placed in the same slot (see Fig 3c & Fig 6).

### 4.4.2  *EKB* supports relational database operations of Permutation And Join

*EKB* is similar to a relational database [50] with Observations as columns, Hypotheses as rows and Tests as values, but with potential associations among values for deduction Knowledges. Relational database operations independent of values associations, such as Permutation (switching rows and columns) and Join (merging *EKB*s) can be implemented without compromise in *EKB*; operations dependent on values associations such as Restriction (select rows), Projection (select columns) and Compositions (merge selected columns&rows from multiple *EKB*s) can only be implemented for *EKB*s storing only induction or abduction Knowledge and have no associations among Tests.

*EKB*s are highly flexible for team collaboration and maintenance. Different team members working on different Containers can share one *EKB* as the common work space to improve efficiency. Multiple *EKB*s can be joined together without information loss so that Knowledges produced by different teams or team members can be accumulated into one *EKB*. For *EKB*s storing only induction and abduction Knowledge, all relational database operations are applicable, so that team members can compose their own *EKB*s without keeping a potentially large team *EKB* on the local machines.

## 5   Advantages and pain points alleviated

Inspired by the philosophy of logical reasoning, *Evident* is intuitive to understand and follow. Project activity and artifact containerization supports incremental as well as adaptive project planning and artifact pattern abstraction. Disentangling Hypotheses and Observations reduces unnecessary dependency, cascade correction and uncertainty in project planning. Knowledge representation in associations among artifacts can not only track Knowledge development, but also Knowledge development status (prove, disproved or overlooked), which improves project measurability, trackability and reproducibility. Overall *Evident* may help applicable projects deliver fast and at scale with many pain points alleviated in DM, ML and KM (see Table 2).

### 5.1   DM

Containerized Data and Models in *Evident* prevents unstable data dependency, model entanglement and cascade correction. Dead data and codes can be easily identified and removed. Standardized Models and Experiments encourage reuse of computationally efficient containers, support automatic DM. Different experiments may use different optimization target function on the same Model and Data to deliver different Knowledges for different users, e.g. Marketing vs Engineering managers.

Containerization is an alternative to the state of the art artifact version control, such as git, which keeps only the one version of the code or data in the workspace with historic versions saved as commits. *Evident* keeps all applicable versions available in the workspace for easy access. This may appear to use more storage space. However current version control tools all save version commits as snapshots [51], demanding comparable storage space of *Evident* if a *Evident* equivalent number of versions are stored.

*Evident* granulates project activities into independent standardized Knowledge and Container levels, supports adaptive development, facilitates project planning among collaborators in teams of various sizes and reduces planning overhead. Artifacts are continuously stored in *EKB*, making project development measurable, trackable, reproducible and scalable. Meanwhile once Containers are produced, Knowledge or documentation reports can be generated automatically instead of manually. *Evident* accelerates DM delivery in both short-term and long-term.

### 5.2   ML

A deployed Algo can be evaluated easily by re-applying the original Research Experiment on the production data. Different users involved in the deployment can understand the Algo's scope and origins easily by examining the research Experiment (see Fig 4b). Research Experiments can load data in real time as Production Experiment, so that both Experiments can inherit the same design patterns with statistical analysis and evaluation metrics. The Production Experiment can report and examine the prediction performance at regular time intervals to detect production data pattern drift for either model reconfiguration or model replacement. Once a model with its configuration is retired from production, the production data is containerized and associated with the Production Experiment, transforming the Production Experiment into Research Experiment and a deduction Knowledge for prediction into an induction Knowledge that is also preserved in *EKB*.

Because both research and production can operate on the same *EKB*, research and production team members can share the same workspace the way software engineers work on the same code repository, facilitating the model migration from research to production and efficient team collaborations.

### 5.3 KM

Knowledge formatting into design patterns of Containers provides a meaningful progress towards Knowledge standardization for improved definition, reference and storage compared to state of art sentences or articles. *EKB* with standardized Container templates may offer potential tools for people to examine the Hypotheses formed against evidence or Observations, facilitating evidence-based decision making and Knowledge development. *EKB* can not only facilitate Knowledge dissemination, accumulation and retention, but also label the development status of each Hypothesis as proved, disproved or overlooked, a desirable design pattern for projects and Knowledge Management.

## 6 Discussions

### 6.1 Significance

*Evident* may advance many society domains such as software, philosophy, science, business as well as Knowledge sharing and retention across the globe, thanks to its applicability to general KM.

*EKB* may make no smaller impacts than relational data base [50], the invention of which created a data base industry, as one solution to store information as Knowledge, a granular level above data.

### 6.2 Work to do

Work needs to be done regarding *Evident* ergodicity over logical reasoning in philosophy. If proved, *Evident* can support all logical reasoning in philosophy. No evidence has existed to prove or disprove the claim. *Evident* ergodicity is overlooked, stated in *Evident* language, especially considering logical reasoning in philosophy may evolve.

Control studies need to be done to show *Evident* can truly provide value. No tools tailored to support *Evident* project development planning and *EKB* are available, although some existing tools are applicable for use. Particularly the tools that allow unexpected alteration proof, easy access and visualization of Containers are in demand. More detailed discussions need to be done about how *Evident* help applicable projects with examples.

### 6.3 Pain points not alleviated

*Evident* offers no detailed guidance in development below Container level. For example, a Model or computation flow cannot be broken down further into smaller modules by *Evident* for incremental and adaptive development. Multiple languages smells and accidents running prototype Algo in production cannot be avoided by *Evident* either. In addition, *Evident* cannot control future observation alteration caused by decisions made by people based on *Evident* produced Knowledge.

*Evident* can only manage project artifacts of data or codes but not sensors or hardwares. Pain points caused in data acquisition such as availability, inaccuracy and inconsistency are out of *Evident*'s scope. *Evident* is incapable of improving computation hardware resources allocation either.

### 6.4 More words about Agile

Due to Agile's ambiguity and unfalsifiability as a scientific claim, it might be a better practice to drop the term Agile and instead quote each framework currently under Agile on its own. Frameworks such as iterative and evolutionary development as well as Kanban are valuable although need to be employed discretionally. Practitioners should have better understood what exactly they were doing without being fuzzed by the buzzword Agile.

## 7 Conclusions

The paper proposes *Evident* as a project development and artifact management methodology for DM, ML and KM as well as *EKB* as a Knowledge base topology. *Evident* and *EKB* have been shown of great value to alleviate many unresolved pain points. *Evident* has the potential to facilitate the advancement of many aspects of society due to its utility in general Knowledge management. *EKB* may serve as the infrastructure for storing information as Knowledge, a granular level above data.

# A Appendix

## A.1 The following among the four Values and twelve Principles of The Manifesto [24] appear to be goals:

Value 4: Responding to change over following a plan;

Principle 1: Customer satisfaction by early and continuous delivery of valuable software.

Principle 2: Welcome changing requirements, even in late development.

Principle 3: Deliver working software frequently (weeks rather than months)

Principle 7: Working software is the primary measure of progress

Principle 8: Sustainable development, able to maintain a constant pace

Principle 9: Continuous attention to technical excellence and good design

Principle 10 : Simplicity—the art of maximizing the amount of work not done—is essential

## A.2 The following in The Manifesto appear to be approaches but vaguely defined:

Value 1: Individuals and interactions over processes and tools

Value 2: Working software over comprehensive documentation

Value 3: Customer collaboration over contract negotiation

Principle 5: Projects are built around motivated individuals, who should be trusted

Principle 11: Best architectures, requirements, and designs emerge from self-organizing teams

## A.3 The following in The Manifesto appear to be actionable approaches but irrelevant of iterative, evolutionary or incremental development regarded as Agile by most people [9]:

Principle 4: Close, daily cooperation between business people and developers

Principle 6: Face-to-face conversation is the best form of communication (co-location)

Principle 12: Regularly, the team reflects on how to become more effective, and adjusts accordingly

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
