# OpenReview forum: "Evident: a Development Methodology and a Knowledge Base Topology for Data Mining, Machine Learning and General Knowledge Management"
_NeurIPS.cc/2022/Conference — NeurIPS 2022 Submitted_

### Official Review · Reviewer_FzW8 · 2022-07-10

**Rating:** 3
**Confidence:** 2
**Soundness:** 2 fair
**Presentation:** 2 fair
**Contribution:** 1 poor

**Summary:**

This paper proposes a project management/artifact management framework (Evident) in the form of a knowledge base topology. Throughout the paper, the authors discuss about the challenges presently faced by the AI community - including in knowledge representation and utilisation of traditional relational databases for ML/AI workflows. The paper is interesting and the problem being solved is relevant.

**Questions:**

This reviewer believes that the paper is very theoretical and oriented more towards the "project management" audience in comparison to the substance which an AI audience would be interested in. It would have been useful if this paper had more interaction with AI - beyond the very common pitfalls ("pain points") that the paper describes which are mostly well known in terms of consensus in the community. Also, without any user-studies or real-world use of this framework, the value which Evident can deliver is very hypothetically presented in the paper which makes it less rigorous. This reviewer suggests that paper would be more suitable for a journal relevant to AI and project management etc. in its current form.

**Limitations:**

Mostly the limitations have been addressed. However, there is not much added value or substance which the paper describes beyond the "Pain points not alleviated" described in Section 6.3 - the hypothetical nature of the Evident framework makes the paper read more like an essay (being very verbose) which affects the overall contributions outlined in the paper.

**Strengths And Weaknesses:**

Weaknesses:
Very limited novelty in terms of the contributions of the paper. While the framework and discussions surrounding Evident are interesting, they do not add much substance from an AI perspective and from the project management perspective, it is not very clear how and why the proposed workflow is the way to go for the AI community. While the authors have described "pain points of DM, ML and KM", it is very theoretical with limited examples, limited practical scenarios and above all, limited interrelation with the substance related to AI over project management.

Strengths:
Interesting topic being discussed in the paper, very well written paper.

---

> ### Author Response · Authors · 2022-08-02
> **We Humbly Disagree That Ideas Have to Be Accompanied with Quantitative Case Studies When Proposed to Make Significant Impact**
>
> (Common Response to All Reveiwers)
>
> The authors appreciate the reviewers’ time and effort trying to make the draft a better paper. We are glad to receive the feedback and have the chance to know how the community looks at our work. We respectfully disagree with the reviewer’s concerns on the paper’s innovation and place our responses as below. Most importantly we listed three ideas of significant contribution without quantitative case studies when proposed in computer science, sociology and political economics. We also listed one example that was missed out of the draft due to space limitation. We will be happy to add the example in with one extra page space if the draft can be accepted. The authors hope to have addressed the reviewer’s concerns and the reviewers can re-evaluate the draft for acceptance for NeurIPS 2022. We will also be happy to discuss with the reviewers during Reviewer-Author Discussion.
>
> (Response to Reviewer FzW8)
>
> Weakness:  We humbly believe ideas do not necessarily have to be accompanied by case studies when proposed. We copy what we wrote to address the similar concern of Reviewer P1e5 as below.
>
> - First and most, ideas can be of great value even without quantitative case studies when proposed. Following are three examples in computer science, sociology and political economics.
>
> a. One example would be the relational database proposed by E.F. Codd in 1970 ([50] in the draft), before when nobody had implemented the idea. However, IBM, Oracles and other companies implemented the idea and developed SQL along with the relational database industry in the next decades. The truth is, the inventor E.F. Codd did not even get the chance to implement his own idea at his employer (IBM) (https://en.wikipedia.org/wiki/Edgar_F._Codd). However, this did not keep the relational database idea he proposed from making significant contributions in the database industry.
>
> b. When Jean-Jacques Rousseau popularized the idea of Social Contract, he did not show any quantitative case studies to prove his idea. However, the idea of Social Contract became the foundation of the American Revolution (eventually the United States), France Revolution and much more.
>
> c. When Adam Smith built up the foundation of modern political economics in The Wealth of Nations, he did not show any quantitative case studies in most of his claims either. However, his contribution is substantial.
>
> - Secondly,  we did integrate a dummy project example during describing Evident methodology and Evident Knowledge Base (EKB). The dummy project includes different knowledge types of different pairs of Observations and Hypotheses (Induction Knowledge, Deduction Knowledge and Abduction Knowledge) in Fig 3 (a) & (b) and of the same pair of Observation and Hypothesis due to different Tests Fig 3 (c). We show how Evident can guide project development and store different types of knowledge into “one” Evident Knowledge Base in Fig 5 and Fig 6. The benefits of Evident and EKB are described in the drafts conceptually
>
> - Thirdly, one co-author implemented one automatic US stock trading and research system based on the methodology of Evident proposed in the draft. He had a full time job which had nothing to do with finance, but still was able to efficiently manage a huge periodic updated data set, research algorithms and put the algorithm into production (automatic trading a decent amount of real money based on minute level data with around twenty or thirty trades per day). He completed four algorithm improvement iterations and traded for around 14 months in total over years with reasonably good performance. Without such an algorithm research and production system guided by Evident, he could never manage such a relatively large project all by himself as a side project. However, the draft has nine full pages with spaces squeezed and unnecessary contents removed for submission. If the reviewer can help point out which content in the draft is unnecessary, we are more than happy to remove the unnecessary content and add this concrete example. Or we can add this example after the draft is accepted with one extra page space for final publication submission.
>
> - Fourthly, our contributions such as listing Agile’s drawback in unfalsifiability and alleviating current DM and ML pain points can be understood conceptually. We already listed control study as future work in Line 292 as the paper’s limitation.
>
> - Lastly, we believe by proposing the ideas of Evident as a project management methodology and EKB as a knowledge base topology, the rest of the community can implement them just as the idea of relational database invented 52 years ago and prove our opinions about Evident either right or wrong.

---

> > ### Author Response · Authors · 2022-08-02
> > **To Continue the Rebuttal**
> >
> > Questions: This reviewer believes that the paper is very theoretical and oriented more towards the "project management" audience in comparison to the substance which an AI audience would be interested in. It would have been useful if this paper had more interaction with AI - beyond the very common pitfalls ("pain points") that the paper describes which are mostly well known in terms of consensus in the community. Also, without any user-studies or real-world use of this framework, the value which Evident can deliver is very hypothetically presented in the paper which makes it less rigorous. This reviewer suggests that paper would be more suitable for a journal relevant to AI and project management etc. in its current form
> >
> > - The authors believe NeurIPS is the right venue for the draft for the following reasons:
> >
> > a. A majority of NeurIPS attendants are DM or ML practitioners who are interested in DM or ML topic. They are also related to computer science, who know Agile pretty well. They are some of target audience of this paper
> >
> > b. EKB is similar to relational database, the prevailing database topology, which ML community is familiar with.
> > Many attendants should be interested in DM/ML project managements.
> >
> > c. We See the possibility of elevating DM and ML into a higher level of KM for science, business, military, education in general. We also see the needs to raise the awareness of DM and ML practitioners to retain and manage knowledge within organization including but not limited to business, investment or military decisions.

---

> > > ### Author Response · Authors · 2022-08-02
> > > **Yann LeCun's Criteria to Review Paper**
> > >
> > > We sincerely hope the reviewers can reconsider the acceptance of our draft based on the criteria Yann Lecunn, Turing Award Winner, uses to review papers:
> > >
> > > "I don't review often, but I'm a pretty gentle reviewer. I'm asking myself "would the community be better off with or without this paper?" (https://twitter.com/ylecun/status/1526038959347970049)
> > >
> > > We do see the value of our draft to the community and hope it can be accepted.

---

### Official Review · Reviewer_Fty3 · 2022-07-11

**Rating:** 3
**Confidence:** 3
**Soundness:** 1 poor
**Presentation:** 2 fair
**Contribution:** 1 poor

**Summary:**

The paper discusses problems (called pain points in the paper) in using the agile methodology for Data mining/Machine learning/knowledge management. It then presents a software development methodology for data mining tasks that can also be used for machine learning and knowledge management. Activities in a project are broken down into observations, hypotheses and tests that denote facts, knowledge and evaluation respectively. These serve as containers that can also used for storing and managing project related artefacts.


**Questions:**

- The authors have discussed the agile methodology and problems in its application for software development and artefact management for dm/ml/km. However, why compare with agile only? It would be better to compare with other DM methodologies that have been proposed. For example, the paper talks about the CRISP-DM, but lists limitations of this methodology briefly. No other methodology is mentioned.

- How have goals and approaches been differentiated? For example, how is Individuals and interactions over processes and tools an approach and Responding to change over following a plan a goal

- The paper says only four principles are actionable, but lists only three

- The fact that agile is unfalsifiable is mentioned, and treated as a drawback. However, it is not clear which development methodologies  (if any) are falsifiable

- The paper says the scope of Agile is not defined. Again,  it would be good to mention whether all other methodologies have scope defined

- How have the pain points in Table 2 been identified? What is the last column?

- How do the  Fig 3 a), b) and c) represent induction, abduction and deduction? Some explanation is required

- Were experiments not carried out using Evident? At least something concrete is required, perhaps a case study with examples. Otherwise, the claims about alleviation of pain points in section 5 are not justified.

- The paper title mentions topology. Where is this discussed subsequently?

- I am not sure I understand what is meant by the following:

"This paper discusses limitations in agile as a scientific claim"
"Beyond software development, Evident is illustrated to be applicable in many aspects of society"
"Although control experiment challenges or absence of quantitative project Agility measurements may explain no measurable scientific evidence,...."
"DM has not been regarded highly collaborative and scalable activities to deliver high throughput Knowledge"
"resulting in the under performance of production Algo"



**Limitations:**

As far as sections are concerned, the authors have mentioned pain points not alleviated under section 6.3.

**Strengths And Weaknesses:**

Strength:

- The paper identifies the need for having a software development methodology for DM/ML/KM
- The paper presents a summary of pain points for DM/ML/KM

Weaknesses:

- The paper lacks scientific rigor. Although some of the ideas are interesting, there is no experimentation/evaluation to substantiate the proposed methodology
- The paper does not present any literature review of the area, that raises questions about whether the work in the paper is indeed a novel contribution

---

> ### Author Response · Authors · 2022-08-02
> **We Humbly Disagree That Ideas Have to Be Accompanied with Quantitative Case Studies When Proposed to Make Significant Impact**
>
> (Common Response to All Reviewers)
>
> The authors appreciate the reviewers’ time and effort trying to make the draft a better paper. We are glad to receive the feedback and have the chance to know how the community looks at our work. We respectfully disagree with the reviewer’s concerns on the paper’s innovation and place our responses as below. Most importantly we listed three ideas of significant contribution without quantitative case studies when proposed in computer science, sociology and political economics. We also listed one example that was missed out of the draft due to space limitation. We will be happy to add the example in with one extra page space if the draft can be accepted. The authors hope to have addressed the reviewer’s concerns and the reviewers can re-evaluate the draft for acceptance for NeurIPS 2022. We will also be happy to discuss with the reviewers during Reviewer-Author Discussion.
>
> (Responses to Reviewer Fty3)
>
> Weakness 1: We humbly believe ideas do not necessarily have to be accompanied by case studies when proposed. We copy what we wrote to address the similar concern of Reviewer P1e5 as below.
>
> - First and most, ideas can be of great value even without quantitative case studies when proposed. Following are three examples in computer science, sociology and political economics.
>
> a. One example would be the relational database proposed by E.F. Codd in 1970 ([50] in the draft), before when nobody had implemented the idea. However, IBM, Oracles and other companies implemented the idea and developed SQL along with the relational database industry in the next decades. The truth is, the inventor E.F. Codd did not even get the chance to implement his own idea at his employer (IBM) (https://en.wikipedia.org/wiki/Edgar_F._Codd). However, this did not keep the relational database idea he proposed from making significant contributions in the database industry.
>
> b. When Jean-Jacques Rousseau popularized the idea of Social Contract, he did not show any quantitative case studies to prove his idea. However, the idea of Social Contract became the foundation of the American Revolution (eventually the United States), France Revolution and much more.
>
> c. When Adam Smith built up the foundation of modern political economics in The Wealth of Nations, he did not show any quantitative case studies in most of his claims either. However, his contribution is substantial.
>
> - Secondly, we did integrate a dummy project example during describing Evident methodology and Evident Knowledge Base (EKB). The dummy project includes different knowledge types of different pairs of Observations and Hypotheses (Induction Knowledge, Deduction Knowledge and Abduction Knowledge) in Fig 3 (a) & (b) and of the same pair of Observation and Hypothesis due to different Tests Fig 3 (c). We show how Evident can guide project development and store different types of knowledge into “one” Evident Knowledge Base in Fig 5 and Fig 6. The benefits of Evident and EKB are described in the drafts conceptually
> - Thirdly, one co-author implemented one automatic US stock trading and research system based on the methodology of Evident proposed in the draft. He had a full time job which had nothing to do with finance, but still was able to efficiently manage a huge periodic updated data set, research algorithms and put the algorithm into production (automatic trading a decent amount of real money based on minute level data with around twenty or thirty trades per day). He completed four algorithm improvement iterations and traded for around 14 months in total over years with reasonably good performance. Without such an algorithm research and production system guided by Evident, he could never manage such a relatively large project all by himself as a side project. However, the draft has nine full pages with spaces squeezed and unnecessary contents removed for submission. If the reviewer can help point out which content in the draft is unnecessary, we are more than happy to remove the unnecessary content and add this concrete example. Or we can add this example after the draft is accepted with one extra page space for final publication submission.
> - Fourthly, our contributions such as listing Agile’s drawback in unfalsifiability and alleviating current DM and ML pain points can be understood conceptually. We already listed control study as future work in Line 292 as the paper’s limitation.
> - Lastly, we believe by proposing the ideas of Evident as a project management methodology and EKB as a knowledge base topology, the rest of the community can implement them just as the idea of relational database invented 52 years ago and prove our opinions about Evident either right or wrong.

---

> > ### Author Response · Authors · 2022-08-02
> > **To Continue the Rebuttal**
> >
> > Weakness 2: We did a rigorous literature review on current DM/ML challenges and existing methodologies, along with the community’s efforts trying to address DM/ML challenges and ML system designs. However, we decided to state the motivation along the storyline of methodology for the following reasons:
> > - This draft is to propose a methodology and a related knowledge base topology
> > - The storyline appears clear and intuitive to the authors and hopefully the readers
> > - The goal of the draft is not to list all related literatures, but to motivate and propose Evident and EKB.
> >
> > Questions:
> >
> > - The authors have discussed the agile methodology and problems in its application for software development and artefact management for dm/ml/km. However, why compare with agile only? It would be better to compare with other DM methodologies that have been proposed. For example, the paper talks about the CRISP-DM, but lists limitations of this methodology briefly. No other methodology is mentioned.
> >
> > a. Firstly We focused on Agile instead of other methodologies, because Agile is currently the most popular methodologies, not only in software engineering, DM/ML ([28, 29] in the draft), but also in project management and business development. We believe it is the most representative of the state of the art.
> >
> > b. Secondly, CRISP-DM (along with its variants) was the only relevant methodology the authors know that was proposed to deal with data mining. However, it appears more of a theoretical framework and has not gotten much traction as stated in the introduction session of the draft. In fact, many practitioners in the DM or ML domain have never heard about it. Therefore the authors do not see much value by spending efforts on it.
> >
> > c. What’s more important, the draft is not trying to say Evident or EKB is better than any existing methodologies. The introduction of a new intuitive and systematic way of understanding and implementing DM, ML and KM can be of meaningful contribution on its own. We even say Evident and EKB can be implemented along with other applicable methodologies (Line 196 of the draft)
> >
> > d. Lastly, we spent almost 2 out of 9 pages in the draft about Agile and other methodologies. It is probably enough, especially the main idea of the draft is to propose a new methodology and knowledge base topology, not to argue against other methodologies. If the reviewer can point out what content in the draft can be removed, we will be glad to add more comments on other methodologies
> >
> > - How have goals and approaches been differentiated? For example, how is Individuals and interactions over processes and tools an approach and Responding to change over following a plan a goal
> >
> > Based on the authors’ understanding: Goal is what team is established to achieve as outcomes, e.g working software, satisfaction of team colleagues. Approach is the way of working to achieve the goal such as individuals over tools, how often people should meet.
> > The authors agree that “Responding to change over following a plan” can be categorized as both a goal and approach. However, it does not change the fact that the original 4 values and 12 principles of the Manifesto of Agile Software Engineering are somehow random and offer no insights about the approaches to implement evolutionary, adaptive or incremental developments. Especially Value 3: “Customer collaboration over contract negotiation” sounds very random in such a document that’s supposed to be about software engineering.
> >
> > - The paper says only four principles are actionable, but lists only three
> >
> > Our apologies. It’s a typo. Should be three.
> >
> > - The fact that agile is unfalsifiable is mentioned, and treated as a drawback. However, it is not clear which development methodologies (if any) are falsifiable
> >
> > Actually many methodologies are falsifiable, such as Test Driven Development, Waterfall, Evolutionary Development and Kanban. It is possible to decide whether people are implementing these methodologies or not.
> >
> > - The paper says the scope of Agile is not defined. Again, it would be good to mention whether all other methodologies have scope defined
> >
> > a. Some methodologies did mention their scope for example CRISP-DM and its variants, however it does not necessarily mean it’s effective.
> >
> > b. Meanwhile Evident as proposed in the draft, clearly states its scope.
> >
> > c. Even though all other methodologies do not mention scope, it does not mean Agile should not point out its scope. Agile is so commonly used in so many areas, people need to know if they can use Agile in their projects.
> >
> > - How have the pain points in Table 2 been identified? What is the last column?
> >
> > a. Table 2 was identified by summarizing and grouping pain points listed by cited literatures.
> >
> > b. Last column was explained by * at the end of the table. It indicates if the pain points can be alleviated by Agile.

---

> > > ### Author Response · Authors · 2022-08-02
> > > **To Continue the Rebuttal**
> > >
> > > - How do the Fig 3 a), b) and c) represent induction, abduction and deduction? Some explanation is required
> > >
> > > Will add explanation if paper can be accepted with one extra page space
> > >
> > > - Were experiments not carried out using Evident? At least something concrete is required, perhaps a case study with examples. Otherwise, the claims about alleviation of pain points in section 5 are not justified.
> > >
> > > Please see our comments address the Weakness 1 above.
> > >
> > > - The paper title mentions topology. Where is this discussed subsequently?
> > >
> > > We introduce Evident Knowledge Base as the topology and described it in Session 4.4 of the draft.
> > >
> > > - I am not sure I understand what is meant by the following:
> > >
> > > "This paper discusses limitations in agile as a scientific claim"
> > >
> > > It means because Agile can not be clearly defined, there is no way to determine if an Agile project is effective or to falsify the effectiveness of Agile, because nobody can know if a project is implementing Agile or not at the first place. Because falsifiability is the standard to determine if a theory is a scientific theory by Karl Popper. “Agile is effective” is not a scientific claim.
> > >
> > > "Beyond software development, Evident is illustrated to be applicable in many aspects of society"
> > >
> > > Because Evident can be used to improve Knowledge Management in general in science, business, philosophy, military, it can improve society in many aspects.
> > >
> > > "Although control experiment challenges or absence of quantitative project Agility measurements may explain no measurable scientific evidence,...."
> > >
> > > It is very difficult to do two exactly same projects: one for control study and one using Agile.
> > >
> > >  "DM has not been regarded highly collaborative and scalable activities to deliver high throughput Knowledge"
> > >
> > > Shoe or car plants can produce a fairly big amount of shoes or cars per unit time. However, the unit output of DM per unit time is relatively small.
> > >
> > > "resulting in the under performance of production Algo"
> > >
> > > Because the production algo is trained using research data of different patterns from the production data, the algo should have been performing better by using research data of the same pattern as production data.

---

> > > > ### Author Response · Authors · 2022-08-02
> > > > **Yann LeCun's Criteria to Review Paper**
> > > >
> > > > We sincerely hope the reviewers can reconsider the acceptance of our draft based on the criteria Yann Lecunn, Turing Award Winner, uses to review papers:
> > > >
> > > > "I don't review often, but I'm a pretty gentle reviewer. I'm asking myself "would the community be better off with or without this paper?" (https://twitter.com/ylecun/status/1526038959347970049)
> > > >
> > > > We do see the value of our draft to the community and hope it can be accepted.

---

### Official Review · Reviewer_P1e5 · 2022-07-11

**Rating:** 2
**Confidence:** 4
**Soundness:** 1 poor
**Presentation:** 2 fair
**Contribution:** 1 poor

**Summary:**

The paper proposes a new software development methodology (Evident) for machine learning, data mining, and knowledge management. It discusses pitfalls of traditional software processes related to the waterfall and agile models and explains why they may not be sufficient for these fields. Further, the paper describes the main concepts related to Evident. The main idea centers around modeling the relationships between observations, hypotheses, and tests. Finally, the paper describes how this methodology may help to alleviate common pain points in ML, DM, and knowledge management.

**Questions:**

- Has this process been applied in practice? Are there any insights to share to this end?

- For future versions of the paper it may be useful to be more specific and highlight for which of the challenges in Table 2, Evident may work better than other methodologies and why.

**Limitations:**

I do not think the authors have really described or mentioned the limitations of their approach. In contrary, the contributions are sometimes hyperbolically described claiming that the process can advance society, philosophy, and science, without any factual evidence.

**Strengths And Weaknesses:**

Strengths:

S1- The paper aims to make a contribution to the important problem of improving development processes for ML and more.

Weaknesses:

W1- It is difficult to judge the effectiveness of the contribution without any user studies with development teams that have applied Evident. At the very least, it would have been useful to see a few case studies and qualitative remarks on how teams could apply this in practice. This would instantiate the presented terminology (knowledges, containers, hypotheses, observations etc) with concrete tasks and terms of a given project.

W2- Often the paper makes strong claim regarding the ineffectiveness of Agile (or other processes like CRISP-DM) and the effectiveness of of Evident but the justifications are mostly abstract and sometimes handwaving potential benefits without a proper either theoretical or rigorous framework to prove these claims.

W3- It is not clear how the authors distilled the presented challenges in Table 2 from related work. How did they decide to include or leave a challenge out of the table? Plus, there is a lot more work on methods and processes for ML by now that could also be considered. Some of the challenges are also very generic and they may as well apply to any software that may not include ML (e.g. dead code paths, Prototype Algo may be accidentally run in production causing damages).

Literature on challenges in ML software development:

Software Engineering for Machine Learning: A Case Study

Software 2.0 https://karpathy.medium.com/software-2-0-a64152b37c35

Machine learning testing: Survey, landscapes and horizons

“Everyone wants to do the model work, not the data work”: Data Cascades in High-Stakes AI

An empirical study of common challenges in developing deep learning applications

---

> ### Author Response · Authors · 2022-08-02
> **We Humbly Disagree That Ideas Have to Be Accompanied with Quantitative Case Studies When Proposed to Make Significant Impact**
>
> (Common Response to All Reviewers)
>
> The authors appreciate the reviewers’ time and effort trying to make the draft a better paper. We are glad to receive the feedback and have the chance to know how the community looks at our work. We respectfully disagree with the reviewer’s concerns on the paper’s innovation and place our responses as below. Most importantly we listed three ideas of significant contribution without quantitative case studies when proposed in computer science, sociology and political economics. We also listed one example that was missed out of the draft due to space limitation. We will be happy to add the example in with one extra page space if the draft can be accepted. The authors hope to have addressed the reviewer’s concerns and the reviewers can re-evaluate the draft for acceptance for NeurIPS 2022. We will also be happy to discuss with the reviewers during Reviewer-Author Discussion.
>
> (Responses to Reviewer P1e5)
>
> Weakness 1: We humbly disagree with the reviewer about the impossibility to judge the draft’s contribution for the following reasons:
> - First and most, ideas can be of great value even without quantitative case studies when proposed. Following are three examples in computer science, sociology and political economics.
>
> a. One example would be the relational database proposed by E.F. Codd in 1970 ([50] in the draft), before when nobody had implemented the idea. However, IBM, Oracles and other companies implemented the idea and developed SQL along with the relational database industry in the next decades. The truth is, the inventor E.F. Codd did not even get the chance to implement his own idea at his employer (IBM) (https://en.wikipedia.org/wiki/Edgar_F._Codd). However, this did not keep the relational database idea he proposed from making significant contributions in the database industry.
>
> b. When Jean-Jacques Rousseau popularized the idea of Social Contract, he did not show any quantitative case studies to prove his idea. However, the idea of Social Contract became the foundation of the American Revolution (eventually the United States), France Revolution and much more.
>
> c. When Adam Smith built up the foundation of modern political economics in The Wealth of Nations, he did not show any quantitative case studies in most of his claims either. However, his contribution is substantial.
>
> - Secondly,  we did integrate a dummy project example during describing Evident methodology and Evident Knowledge Base (EKB). The dummy project includes different knowledge types of different pairs of Observations and Hypotheses (Induction Knowledge, Deduction Knowledge and Abduction Knowledge) in Fig 3 (a) & (b) and of the same pair of Observation and Hypothesis due to different Tests Fig 3 (c). We show how Evident can guide project development and store different types of knowledge into “one” Evident Knowledge Base in Fig 5 and Fig 6. The benefits of Evident and EKB are described in the drafts conceptually
> - Thirdly, one co-author implemented one automatic US stock trading and research system based on the methodology of Evident proposed in the draft. He had a full time job which had nothing to do with finance, but still was able to efficiently manage a huge periodic updated data set, research algorithms and put the algorithm into production (automatic trading a decent amount of real money based on minute level data with around twenty or thirty trades per day). He completed four algorithm improvement iterations and traded for around 14 months in total over years with reasonably good performance. Without such an algorithm research and production system guided by Evident, he could never manage such a relatively large project all by himself as a side project. However, the draft has nine full pages with spaces squeezed and unnecessary contents removed for submission. If the reviewer can help point out which content in the draft is unnecessary, we are more than happy to remove the unnecessary content and add this concrete example. Or we can add this example after the draft is accepted with one extra page space for final publication submission.
> - Fourthly, our contributions such as listing Agile’s drawback in unfalsifiability and alleviating current DM and ML pain points can be understood conceptually. We already listed control study as future work in Line 292 as the paper’s limitation.
> - Lastly, we believe by proposing the ideas of Evident as a project management methodology and EKB as a knowledge base topology, the rest of the community can implement them just as the idea of relational database invented 52 years ago and prove our opinions about Evident either right or wrong.

---

> > ### Author Response · Authors · 2022-08-02
> > **To continue rebuttal to Reviewer P1e5**
> >
> > Weakness 2: We humbly disagree with the reviewer about our claims against Agile or other methodologies in lack of support.
> > - Firstly, we actually employed a rigorous theoretic framework of Karl Popper using falsifiability to define science to support our claim. We focus on claiming against Agile for its ambiguity and unfalsifiability. If Agile can not be defined clearly, it is impossible to prove it is ineffective or effective as stated in the draft. We believe it is enough to make our point in the draft Line 307.
> > - Secondly, if the reviewer can point out which of our arguments against Agile are defected, we will be more than happy to address.
> > - Thirdly we focused on Agile instead of other methodologies, because Agile is currently the most popular methodology, not only in software engineering, DM/ML ([28, 29] in the draft), but also in project management and business development. We believe it is the most representative of the state of the art.
> > - Lastly, we spent almost 1 out of 9 pages in the draft about Agile. It is probably enough, especially the main idea of the draft is to propose a new methodology and knowledge base topology, not to argue against Agile. If the reviewer can point out what content in the draft can be removed, we will be glad to add more justifications for our claim
> >
> > Weakness 3: We respectfully disagree with the reviewer about rejecting our draft because of how we presented current challenges in Data Mining and Machine Learning.
> > - First of all, this is not a review paper about existing challenges in the domain. The goal is never to be comprehensive. We did miss some related papers as pointed out by the reviewer and are very happy to  include them in. However, we wonder if any paper can be comprehensive listing all related literatures in the domain.
> > - We searched literatures and listed those we believe are representative. After grouping similar challenges together, we organized and listed them in the draft. If the reviewer can point out which ones in the cited literatures are missed, we are more than happy to fix.
> > - The example listed by the reviewer that may not include ML (e.g. dead code path) is actually cited in page 5 of literature [40] in the draft.
> >
> > Question1: Has this process been applied in practice? Are there any insights to share to this end?
> > - Yes, we introduce our experience implementing Evident when addressing the Weakness 1 above. We implemented Evident in a fairly complicated ML research and production project by a team of one member. We have not implemented Evident and EKB in a bigger scale. However, we see their potential not only for Data Mining, Machine Learning for bigger teams, but also for knowledge management in general for science, education, philosophy and so on. We hope the community can implement it and prove us right or wrong.
> >
> > Question 2: For future versions of the paper it may be useful to be more specific and highlight for which of the challenges in Table 2, Evident may work better than other methodologies and why.
> > - As stated in the draft introduction, most other methodologies are proposed during the period when softwares were developed for scientific calculation or computer applications. They were not applicable for software development to discover knowledge through data or knowledge management in general.
> > - CRISP-DM (along with its variants) was the only methodology the authors know that was proposed to deal with data mining. However, it appears more of a theoretical framework and has not gotten much traction as stated in the draft. It is somehow difficult to say how people can implement it. In fact, many practitioners in the DM or ML domain have never heard about it. Therefore the authors do not see much value by doing the comparisons.
> > - What’s more important, the draft is not trying to say Evident or EKB is better than any existing methodologies. The introduction of a new intuitive and systematic way of understanding and implementing DM, ML and KM can be of meaningful contribution on its own. We even say Evident and EKB can be implemented along with other applicable methodologies (Line 196 of the draft)

---

> > > ### Author Response · Authors · 2022-08-02
> > > **Yann LeCun's Criteria to Review Paper**
> > >
> > > We sincerely hope the reviewers can reconsider the acceptance of our draft based on the criteria Yann Lecunn, Turing Award Winner, uses to review papers:
> > >
> > > "I don't review often, but I'm a pretty gentle reviewer. I'm asking myself "would the community be better off with or without this paper?" (https://twitter.com/ylecun/status/1526038959347970049)
> > >
> > > We do see the value of our draft to the community and hope it can be accepted.

---

### Public Comment · ~Mingwu_Gao1 · 2022-11-23
**Paper has been published in arxiv**

Here is the link. Thanks everybody.

https://arxiv.org/abs/2211.10291

---

### Meta-Review · Area_Chair_8jRB · 2022-08-26

**Recommendation:** Reject
**Confidence:** Certain

**Metareview:**

The paper identifies relevant issues in the current software development process for machine learning, data mining, and knowledge management, however it does not provide any practical evidence that the proposed directions can solve the identified issues. The content of the paper is more suited for a position paper than for a technical scientific paper, which is the target of NeurIPS. So, while recognising some value in the contribution of the paper, I believe its nature does not completely match the NeurIPS expectations, and it would be more suitable for conferences where position papers are one of the components of the technical program.

**Award:**

No

---

### Decision · Program_Chairs · 2022-09-14

Reject